# Novel strains of *Campylobacter* cause diarrheal outbreak in Rhesus macaques (*Macaca mulatta*) of Kathmandu Valley

Rajindra Napit[1,2]‡*, Prajwol Manandhar[1,2], Ajit Poudel[1,2], Pragun G. Rajbhandari[1]‡, Sarah Watson[3], Sapana Shakya[1], Saman M. Pradhan[1,2], Ajay N. Sharma[1], Ashok Chaudhary[1], Christine K. Johnson[4], Jonna K. Mazet[4], Dibesh Karmacharya[1,2,5]*

1 One Health Research Division, Center for Molecular Dynamics Nepal, Kathmandu, Nepal, 2 Research Division, BIOVAC Nepal Pvt. Ltd., Nala, Banepa, Nepal, 3 London School of Hygiene and Tropical Medicine, London, United Kingdom, 4 One Health Institute, School of Veterinary Medicine, University of California, Davis, CA, United States of America, 5 The School of Biological Sciences, University of Queensland, Queensland, Australia

‡ These authors contributed equally to this work.
* dibesh@cmdn.org, dibesh@biovacnepal.com (DK); rajindra04@gmail.com (RN)

## Abstract

*Campylobacter spp.* is often underreported and underrated bacteria that present real health risks to both humans and animals, including non-human primates. It is a commensal micro-organism of gastrointestinal tract known to cause gastroenteritis in humans. Commonly found in many wild animals including non-human primates (monkeys- Rhesus macaques) these pathogens are known to be a common cause of diarrhea in humans in many parts of developing and under developed countries. Rhesus macaques from the two holy sites in Kathmandu (Pashupati and Swoyambhu) were included in this cross-sectional study. Diarrheal samples of monkeys were analyzed to detect and characterize the pathogen using 16S rRNA-based PCR screening, followed by DNA sequencing and phylogenetic analysis. Out of a total 67 collected diarrheal samples, *Campylobacter spp.* were detected in the majority of the samples (n = 64; 96%). DNA sequences of the amplified PCR products were successfully obtained from 13 samples. Phylogenetic analysis identified *Candidatus Campylobacter infans* (n = 10, Kimura-2 parameter (K2P) pairwise distance values of 0.002287). Remaining three sequences might potentially belong to a novel Campylobacter species/sub-species- closely relating to known species of *C. helviticus* (K2P pairwise distance of 0.0267). Both *Candidatus Campylobacter infans* and *C. helvitucus* are known to infect humans and animals. Additionally, we also detected the bacteria in water and soil samples from the sites. *Campylobacter spp.* caused the 2018 diarrhea outbreak in Rhesus macaques in the Kathmandu valley. *Campylobacter* might be one of the important contributing pathogens in diarrheal outbreaks-both in humans and animals (monkeys) in Nepal. Due to close interactions of these animals with humans and other animals, One Health approach might be the most effective way to prevent and mitigate the threat posed by this pathogen.

**Data Availability Statement:** DNA sequences are available in the NCBI Genbank with accession number MZ068100 to MZ068112. All the data are

included in the paper including supplementary information.

**Funding:** The author(s) received no specific funding for this work.

**Competing interests:** The authors have declared that no competing interests exist.

## Introduction

*Campylobacter* are gram-negative bacteria that is primarily found in gastrointestinal tract of various species [1]. As of 2022, its genus consists of a diverse group of bacteria comprising of 39 species, 11 sub-species, and 4 biovars [2]. *Campylobacter spp.* is a zoonotic pathogen that is found in the gut flora of many species ranging from domesticated to wild animals- both free roaming and in captivity. It is capable of infecting humans as well as non-human primates (NHP) causing mild to severe gastrointestinal problems [3–5]. Emergence of antibiotic-resistant strains of *Campylobacter* is a major public health concern as animals carrying the bacteria pose a significant risk to humans via contamination of water sources, food, or through repeated interactions (physical contacts) [4–6]. In Nepal, *Campylobacter* is one of the leading causes of food-borne infections; and antibiotic resistant strains of the bacteria have also been reported in poultry slaughter houses throughout Nepal [7–10]. Most of the studies conducted in the country have mostly been limited to food-borne pathogenesis of *Campylobacter* [7–11]. Although, zoonotic spillover is highly prevalent from interactions with animals either carrying or infected with *Campylobacter*, very limited studies have been conducted in Nepal [8]. Limited publications are available documenting spillover from domesticated dogs *Canis lupus familiaris* [12] and from livestock to farmers [13]. However, no such studies have been published highlighting detection and possible spillover of *Campylobacter* from NHP such as Rhesus macaques *(Macaca mullata, commonly known as monkeys)* to humans in Nepal, even though, risk of such exposure and disease transmission have been widely documented [4–6].

*Campylobacter spp.* has been found in both captive and free-roaming monkeys causing diarrhea and severe enterocolitis [4, 14, 15]. Due to NHP's high genomic similarities and close evolutionary relationships to humans, and similar gut flora detected in developing countries, the risk of zoonotic transfer of pathogenic strains of bacteria like *Campylobacter* is highly probable [16, 17]. Previous studies have detected presence of the bacteria in captive NHPs in the United States [18], New Zealand [19], and in Kenya [15]. Presence of pathogenic strains of the bacteria in healthy and asymptomatic monkeys [3] poses even higher risk of zoonotic spillover through direct physical contact or indirectly as a potential source of environmental contamination by fecal matter [20].

In Kathmandu, monkeys inhabit few of the major holy sites including Pashupati and Swoyambhu. These sites, are surrounded by dense urban human population and have significant human-wildlife (monkey) interactions. Following report of diarrhea outbreak (2018) in monkeys of these two sites, we carried out an opportunistic cross-sectional research- specifically designed to detect and characterize *Campylobacter* infections.

Traditionally, *Campylobacter* is detected using microaerophilic culture media (modified charcoal-cefoperazone-deoxycholate agar or mCCDA) [21] but due to the challenge of culturing the bacteria and possibility of getting false-negatives [22], we opted for molecular and sequencing methods instead and present a varied and simplified approach to whole genome sequencing (WGS) as well for strain level identification.

## Materials and methods

### Ethical statement

The research was conducted as a supplementary study to the PREDICT project which focused on understanding emerging diseases in urban-wildlife interfaces. All the permits and ethical clearance were obtained before the study. The non-invasive sampling and analysis were covered by the permit obtained from the Nepal Health Research Council (NHRC, Ref.no. 224).

### Study design and site selection

A cross-sectional study was conducted during active diarrheal outbreaks at two Rhesus macaque inhabiting sites (Pashupati and Swoyambhu) in Kathmandu (Nepal) in 2018 (June -July) [23]. The sampling sites were chosen according to the confined habitat, with a focus on areas with frequent monkey-human interactions. These two holy (temple) sites in the Kathmandu have many free-roaming monkeys and are frequently visited by people for sightseeing and religious purposes. Neighboring areas of the temples in both the sites also have a significant numbers of cattle, dogs, chickens and other birds present.

The Swoyambhunath temple, one of the oldest Buddhist holy sites in the region, is situated on top of a hillock in the northwest of the Kathmandu Valley (Fig 1). Also known as the "Monkey Temple", the area surrounding the Swoyambhunath (with area of 2.5 square kilometer) is home to one of the largest populations of free-roaming macaques in the region. The site hosts an estimated population of 400 monkeys [24]. The Pashupatinath temple is one of the important and popular holy Hindu sites in Nepal. This site is a home to a population of 300 monkeys [24, 25], which reside in nearby patches of the forests (Bankali, Bhandarkhal and Mrigasthali) surrounding the temple premises (Fig 1) [26].

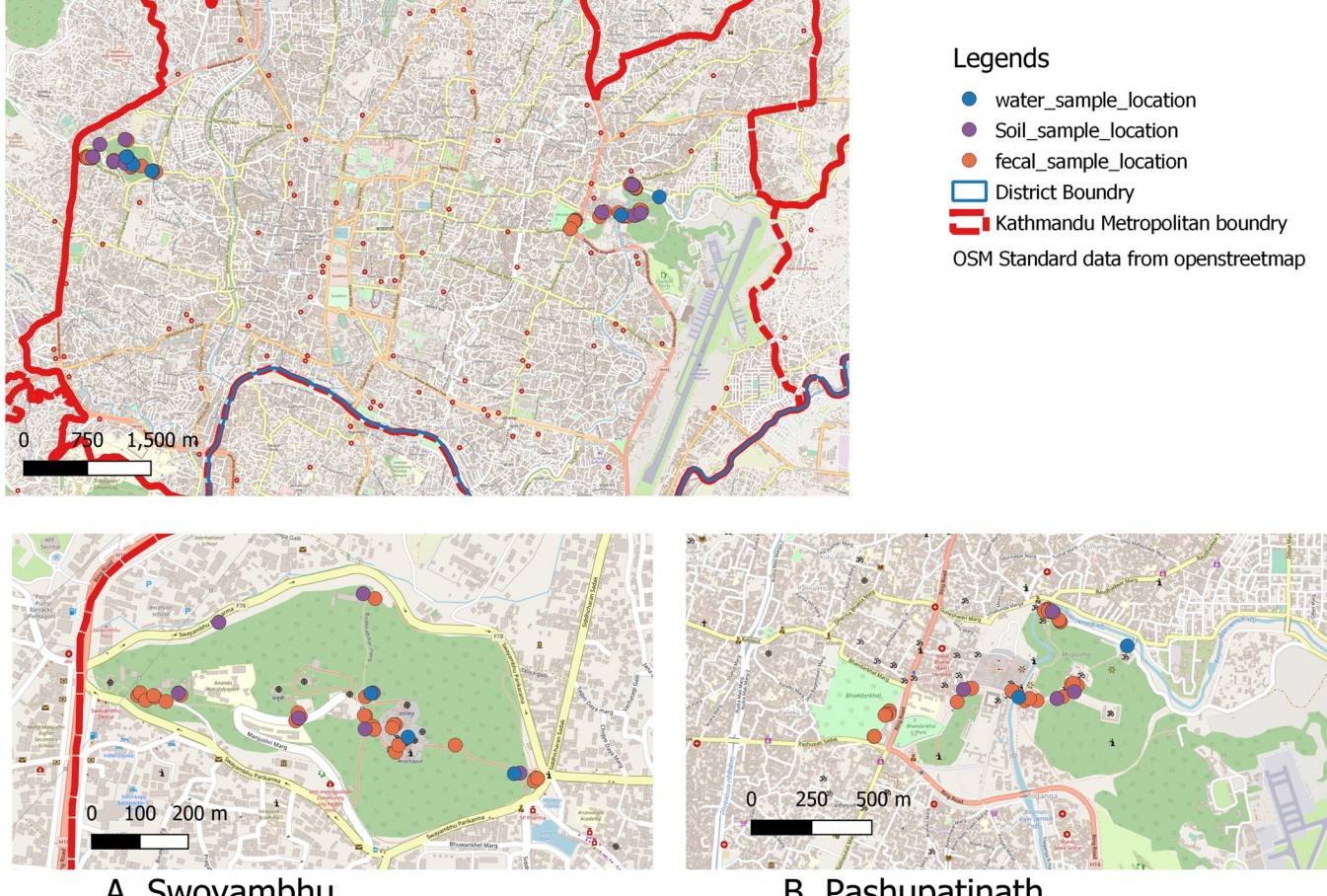

A. Swoyambhu                                    B. Pashupatinath

**Fig 1.** Rhesus macaque diarrheal outbreak sites (A. Swoyambhu B. Pashupatinath) located within Kathmandu Metropolitan city, Nepal (Image was created using QGIS, base map file was obtained from data from OpenStreetMap (OpenStreetMap contributors) [27]) and Nepal administrative shape files obtained from Open data Nepal (opendatanepal.com) [28].

Both the sites were divided into 5 sampling transects, and 5 field teams were mobilized to systematically comb through each of the transect, and collect only diarrheal (loose) fecal samples from monkeys. Feces were collected using sterile swabs in a tube containing phosphate-buffered saline (PBS). A replicate portion of the feces were also collected in silica gel tubes. Additionally, adjacent soil and water samples from a drinking water source (pond at Swoyambhu /river at Pashupatinath) were also collected to check for any possible environmental contamination. The samples were immediately transported in cold-chain to the Intrepid Nepal Lab in Kathmandu and stored at -20˚C freezer for further processing. A total of 67 opportunistic diarrheal fecal samples and soil samples (n = 11) were collected from both the sites. Water samples (n = 5, pond/river and drinking water sources) were also collected (Fig 1).

## Campylobacter detection and characterization

**Molecular screening and 16S rRNA gene sequencing.** Bacterial DNA was extracted from the collected samples (fecal, soil and water) using Bacterial DNA extraction kit (Zymo Research, USA) using manufacturer's protocol. *Campylobacter* was detected by PCR amplifying ~ 800 bp fragment of the 16S rRNA gene using *Campylobacter* genus specific PCR primer sets C412F and C1288R [29]. PCR amplification was done in 25μl volume containing reaction buffer, 0.2nmol primers, Taq polymerase and 2 μl template. The cycling condition for the PCR- initial denaturation at 95˚C for 4 minutes, 35 cycles of denaturation at 95˚C for 30 sec/cycle, annealing temperature of 55˚C for 30 sec and extension at 72˚C for 30 sec, and a final extension at 72˚C for 10 min. The PCR products were separated on 1.5% agarose gel electrophoresis.

8μl amplified PCR product was cleaned using 2μl of ExoSAP-IT™ kit (Thermofisher, Catalog No. 78200.200.UL). The reaction mixture was then incubated for 30 minutes at 55˚C to get rid of excess PCR primers, followed by 85˚C for 10 minutes for reaction deactivation. The purified PCR products were then sequenced on an ABI thermocycler using BigDye™ Terminator V3.1 Cycle Sequencing Kit (Catalog No. 4337455). Excess salts and dye terminators were removed using BigDye® XTerminator™ Purification Kit (Catalog No.4376486). The samples were then analyzed on ABI 310 Genetic Analyzer.

## Phylogenetic analysis

The phylogenetic analysis was performed using BEAST v2.6.4 of partial 16S rRNA sequence [30]. A ~649bp (final size after quality trim) sequences from the NCBI Genbank database was obtained of all known *Campylobacter* species [31]. Along with the sequences obtained from the NCBI database, 13 sequences (Genbank acc: MZ06810 to MZ068112) obtained from this study was also included to form a dataset of 16S rRNA partial sequences (S2 Table in S1 File). All the sequences were aligned using MUSCLE v3.8.425 [32] and was visualized in AliView v1.27 [33]. Model test for BEAST analysis was performed using Bmodel test v1.2.1 [34] for substitution model for 10 million iterations. The phylogenetic tree was prepared on BEAST v2.6.4 with HKY substitution model and YuleModel with 100 million iterations, every 1000 trees subsampling and discarding 25% of samples as burn-in. The log file from BEAST was analyzed using Tracer v1.7.2 [35] to verify all the parameter has effective sampling size (ESS) above 200 and tree was visualized/edited using Figtree v 1.4.4 [36]. Further, the mean genetic distance between our 13 unknown *Campylobacter spp*. samples and other *Campylobacter spp*. found in the neighboring clades from the phylogenetic analysis, were estimated through the Kimura 2-parameter (K2P) distance measure using MEGA 11 v11.0.10 [37].

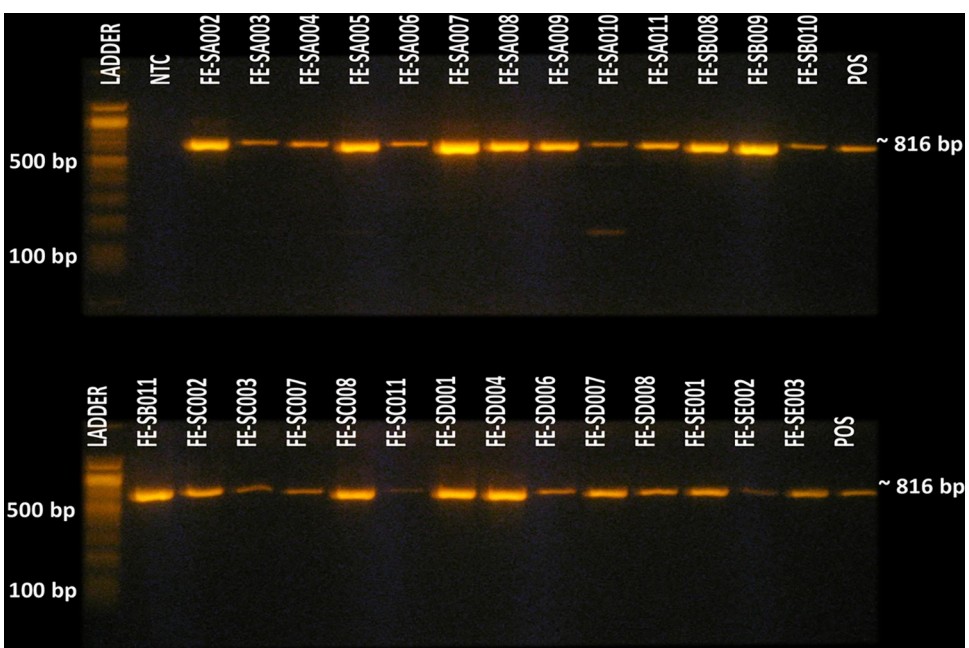

**Fig 2. Detection of Campylobacter sp. in rhesus macaque fecal samples using PCR.**

## Results

Out of the 67 fecal samples, 64 (95.5%) had *Campylobacter* (Fig 2). Some soil (n = 1) and water (n = 1) samples were also contaminated with the bacteria (Table 1).

### Sequencing and phylogenetic analysis

Out of the 64 samples, only 13 provided acceptable quality of 16S DNA sequences, which was used to conduct phylogenetic analysis to resolve taxonomy of the detected *Campylobacter* isolates.

The topology of the phylogenetic tree constructed showed that different species of *Campylobacter spp*. clustered in distinct clades. The isolates having same host species grouped together in a same clade with some exceptions. The isolates clustered into distinct two clades identified as Clade-1 and Clade-2 (Fig 3). The clade-2 samples (PE005, PB002 and SA003) clustered together with *C. upsaliensis*, *C. vulpis*, *C. helveticus*, *C. troglodytes*. Whereas clade-1 (SA002, SA004 –SA009, SA011, SB008 & PD003) samples clustered together in a monophyletic clade comprising of a recently discovered species- *Candidatus Campylobacter infans*.

The Kimura 2-paramater (K2P) pairwise distance between Clade-1 isolates and the recently discovered species of the *Candidatus Campylobacter infans* (Genbank acc: CP049075) was 0.002287 after averaging the pairwise distances (S2 Fig in S2 File). Similarly, the average K2P pairwise distance between Clade-1 isolates with an isolate of *C. hyointestinalis subsp. lawsonii*

**Table 1. *Campylobacter* in fecal samples of rhesus macaque, soil and water from two sites in Kathmandu.**

| | Total number of sample | Number (%) *Campylobacter spp.* present |
| --- | --- | --- |
| Rhesus Macaque Fecal | 67 | 64 (95.5%) |
| Water | 5 | 1 (20%) |
| Soil | 11 | 1 (0.9%) |

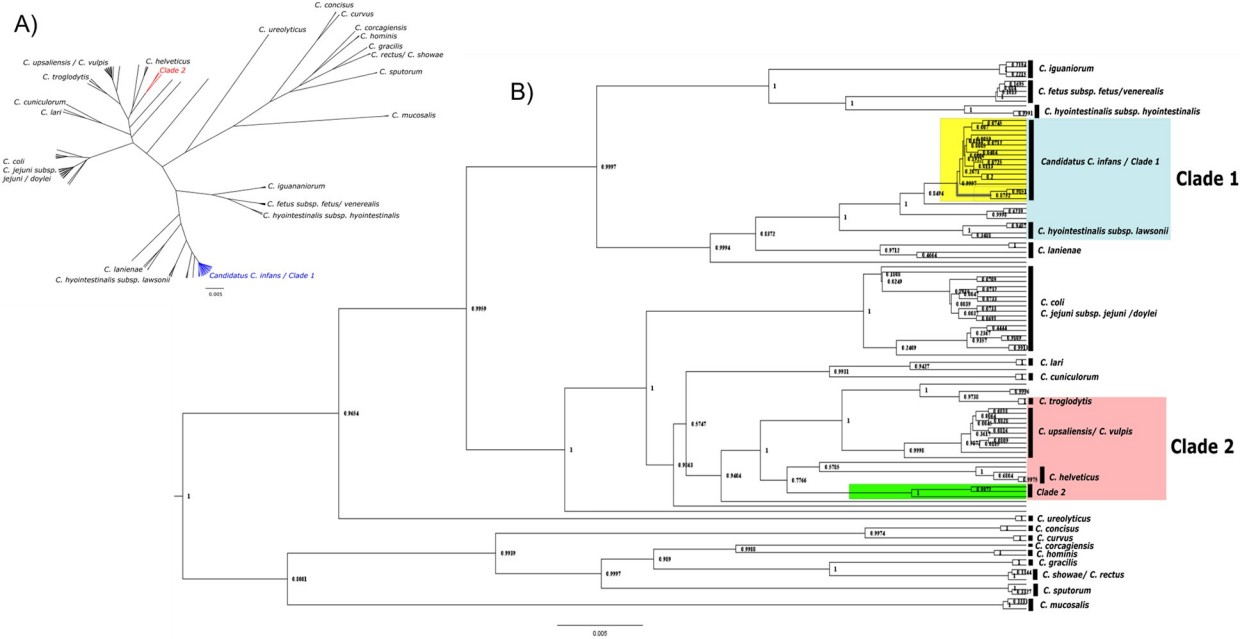

**Fig 3. Phylogenetic tree of *Campylobacter spp*. showing all the detected and reference sequences [constructed using Bayesian inference (BEAST v 2.6.4) and visualized/edited using Figtree v 1.4.4].** A) Cladogram of *Campylobacter spp*. B) Phylogenetic tree constructed using detected and reference *Campylobacter spp*.

(Genbank acc: HQ628645) found in the neighboring clade to the isolates, was calculated to be 0.0206. The average K2P pairwise distance of the Clade-2 isolates was compared with a representative isolate of monophyletic clades of *C. upsaliensis* (Genbank acc: AF550642), *C. troglodytes* (Genbank acc: EU559331) and *C. helveticus* (Genbank acc: DQ174161) and was calculated to be 0.033, 0.0285, and 0.0267, respectively (S3 Fig in S2 File).

## Discussion

Zoonotic pathogens are one of the most common sources of emerging diseases [38]. Campylobacter is considered to be one of the most prevalent zoonotic pathogens [39] that might be contributing to a broader antimicrobial resistance, especially in low and middle-income countries [40, 41]. Nepal has a very high burden of *Campylobacter* infections [7, 9, 42] and it is implicated for causing diarrhea in some of the international travelers; it is also known to cause acute gastroenteritis in children in Nepal [43–45]. Rhesus macaques (monkeys) have close-interactions with humans and other domestic animals, including dogs, in some areas of densely populated Kathmandu. The risk of zoonosis and reverse zoonosis of disease between monkeys and humans are high [39].

The detection and diagnosis of *Campylobacter* infection has been challenging due to inefficiency in widely used culture-based detection [46]. However, availability of molecular based diagnostic techniques have proved to be effective in detecting bacterial species that are normally difficult to culture [47, 48]. Although culture methods are more feasible compared to molecular and sequencing methods, and selective culture can provide qualitative diagnosis comparable to PCR, it only identifies up to the genus level, usually takes almost a day longer compared to PCR, and *Campylobacter's* low metabolic activity and often non-culturable characteristics under unfavorable conditions can also lead to false-negative results [22, 49]. Multiple studies have shown molecular methods having significantly higher sensitivity and

specificity [22, 49, 50] and is often preferred when presence and burden along with strain identification of the disease is required [49]. In this study, we were interested in identification of *Campylobacter* at species level, thus, molecular and sequencing methods were preferred. This might be the first study in Nepal where *Campylobacter* is directly detected in macaque feces. The *Campylobacter* was detected in water and soil samples as well, which increases pathogen spillover possibility amongst various species intermingling at the sites [51].

The phylogenetic analysis of 16S rRNA gene sequences of *Campylobacter* showed presence of two clades of the bacteria (Fig 3). The phylogenetic tree in our study also revealed more or less identical (S4 Fig in S2 File) structure of bacteria clades as found in the study published by Wilkinson D. A., et.al. (2018) [52]. That study used more elaborate whole genome sequence data, and the fact that our study also produced similar *Campylobacter* clade structure using only 16S rRNA fragment sequence data, validates the utility and accuracy of our technique.

The Clade-1 isolates clustered together with the newly reported species of *Campylobacter* (*Candidatus C. infans*), though neighboring clades consisted of other species as well (Fig 3). However, these isolates of other species in the Clade-1 could have been misidentified by the authors, as the K2P distances of isolates found in Clade-1 have close distances to *Candidatus C. infans*, (S2 Fig in S2 File) inferring that all isolates in the Clade-1 could possibly be identified as this newly discovered species. Previous cases of *Candidatus C. infans* have been isolated from infant in the sub-Saharan Africa and South Asia, and from a human sample in the Netherlands [53]. All the *Campylobacter spp*. isolated in this study originated from monkey fecal samples, raising a serious concern of spillover to other species including humans [54, 55]. Furthermore, an isolate from India found in the Clade-1 was isolated from sheep alluding to strong evidence to support zoonotic plasticity.The Clade-2 contained three remaining sequences from this study. The clustering of C. *helveticus*, *C. upsaliensis* and *C. troglodytes* in a nearby clades could also be observed as sister clades to the Clade-2 isolates. Our three isolates formed separate monophyletic clades (Fig 3) within the Clade-2 indicating the findings of a probable new species or sub-species of the *Campylobacter*, which was further supported by the K2P pairwise distance with the closest distance of 0.0267 being with *C. helveticus*, which is similar to the values of the Clade-1 isolates against *C. hyointestinalis subsp. lawsonii* (HQ628645). However, further investigation with a longer 16S rRNA gene fragment or whole genome sequences may be required to properly verify the inferred result.

Samples were collected from two different sites separated by dense human settlements-almost making them wooded islands within the urban landscape. The *Campylobacter* isolates from both the sites clustered together in either one of the identified clades (Fig 3). This result suggests there might be some complex interactions taking place between these two animal populations. Since monkeys from these two sites rarely intermingle, the disease spread might be limited as well. However, humans might help spread *Campylobacter* between these two populations of monkeys.

*C. hyointestinalis* is a commensal organism of pigs whereas *C. helveticus* is commensal in dogs and cats [52]. *Campylobacter* can potentially infect monkeys from these animals, and/or through humans as intermediate hosts. Considering other species such as birds, cats and dogs are also interacting closely with the monkeys at these two sites, the *Campylobacter* reservoir, spillover and transmission are truly playing out in One Health dynamics. Hence, this study highlights the importance of One Health approach to understand and prevent emerging, re-emerging and prevalent infectious diseases.

Diarrheal diseases are one of the most devastating public health concerns in Nepal, especially in a densely populated metropolitan cities like Kathmandu. *Campylobacter* might be one of the important contributing pathogens in diarrheal outbreaks-both in humans and animals (monkeys). We hope that our study will draw attention to this problem and help public health

experts in formulating a plan to cure and prevent this kind of outbreak in macaques, thereby, preventing spillover to humans.

## Supporting information

**S1 File. All the supplementary tables containing additional data.**
(DOCX)

**S2 File. All the supplementary Figures containing additional data.**
(DOCX)

**S1 Raw images. Raw gel images.**
(PDF)

## Acknowledgments

We would like to thank the Pashupati Area Development Trust and the Federation of Swoyambhu Management and Conservation for providing the permit and facilitating this research. We would also like to thank PREDICT Consortium for their assistance during various phases of the project. We would also like to show our gratitude to the Metropolitan City of Kathmandu for helping us with the study. Finally, we thank all the staffs of Intrepid Nepal including Biswo Parakram Shrestha, Samita Raut, Dhiraj Puri for assisting in field sampling and lab experiments during the course of this study.

## Author Contributions

**Conceptualization:** Rajindra Napit, Ajay N. Sharma, Dibesh Karmacharya.

**Data curation:** Rajindra Napit, Prajwol Manandhar, Pragun G. Rajbhandari, Sarah Watson, Saman M. Pradhan, Ashok Chaudhary.

**Formal analysis:** Rajindra Napit, Prajwol Manandhar, Ajit Poudel, Pragun G. Rajbhandari, Sapana Shakya.

**Funding acquisition:** Christine K. Johnson, Dibesh Karmacharya.

**Investigation:** Rajindra Napit, Sarah Watson, Ashok Chaudhary.

**Methodology:** Rajindra Napit, Pragun G. Rajbhandari, Sarah Watson, Saman M. Pradhan, Ajay N. Sharma, Ashok Chaudhary.

**Project administration:** Rajindra Napit, Sarah Watson, Saman M. Pradhan, Ajay N. Sharma, Dibesh Karmacharya.

**Resources:** Christine K. Johnson, Jonna K. Mazet, Dibesh Karmacharya.

**Software:** Rajindra Napit, Prajwol Manandhar, Pragun G. Rajbhandari.

**Supervision:** Christine K. Johnson, Jonna K. Mazet, Dibesh Karmacharya.

**Validation:** Rajindra Napit, Prajwol Manandhar, Ajit Poudel, Sarah Watson.

**Visualization:** Rajindra Napit, Prajwol Manandhar, Ajit Poudel, Pragun G. Rajbhandari.

**Writing – original draft:** Rajindra Napit, Ajit Poudel, Pragun G. Rajbhandari, Sapana Shakya, Dibesh Karmacharya.

**Writing – review & editing:** Rajindra Napit, Prajwol Manandhar, Ajit Poudel, Jonna K. Mazet, Dibesh Karmacharya.

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
