## [Decision Letter · Decision Letter 0]

21 Oct 2022

PONE-D-22-17357Novel strains of Campylobacter cause diarrheal outbreak in Rhesus macaques (Macaca mulatta) of Kathmandu ValleyPLOS ONE

Dear Dr. Karmacharya,

Thank you for submitting your manuscript to PLOS ONE. After careful consideration, we feel that it has merit but does not fully meet PLOS ONE’s publication criteria as it currently stands. Therefore, we invite you to submit a revised version of the manuscript that addresses the points raised during the review process.

We look forward to receiving your revised manuscript.

Kind regards,

Dwij Raj Bhatta, PhD

Academic Editor

PLOS ONE

Journal Requirements:

Additional Editor Comments:

This important research on Campylobacter spp. and role in epidemic diarrhoea in Primates needs minor revision! The authors are suggested to answer queries of reviewers& adress comments from editor! In the abstract section highlighted text is confusing, sentence be changed! commensal flora of which animal species ??chicken?

In introduction part:mention types of standard cultural methods and biochemical identification ,confirmation of these microaerophilic bacteria briefly with proper reference ! Whether the molecular methods applied in the present research is enough to claim novel species? Why cultural methods not applied first to identify isolates and followed by Sequencing of such isolates?

Reviewers' comments:

Reviewer's Responses to Questions

**Comments to the Author**

1. Is the manuscript technically sound, and do the data support the conclusions?

Reviewer #1: Yes

Reviewer #2: Yes

2. Has the statistical analysis been performed appropriately and rigorously? 

Reviewer #1: Yes

Reviewer #2: I Don't Know

3. Have the authors made all data underlying the findings in their manuscript fully available?

Reviewer #1: Yes

Reviewer #2: Yes

4. Is the manuscript presented in an intelligible fashion and written in standard English?

Reviewer #1: Yes

Reviewer #2: Yes

5. Review Comments to the Author

Reviewer #1: This study reports the novel strains of Campylobacter causing diarrhoeal outbreaks in monkeys in two holy sites of Kathmandu valley.

Major comments:

1.In abstract, the authors have mentioned “Opportunistic diarrheal samples of monkeys were analyzed to ………………” It is better to write diarrhoeal samples only instead of opportunistic samples.

2.In abstract, your result highlights the detection of Campylobacter in soil and water samples but methodology part is missing. Please mention about collection and processing of soil and water samples

3. In methods section, line 99, the authors have mentioned water samples were collected form ponds. In contrast to this, in line 102, they have mentioned 5 water samples from river and drinking water sources. Please write exactly which samples you have collected. Also elaborate on the collection and transportation of water samples.

4.In figure 1, please show the sample collection sites for water and soil samples

Minor comments:

Please correct the grammatical errors in the text such as

1.Line 81: Please rearrange the sentence.

2.Line 100: Please mention the name of the laboratory instead of our lab.

3.Line 111: Please write the symbol µ instead of u

4.Line 111-113: Please complete the sentence “ The cycling…….4°C.”

Reviewer #2: The manuscript is well written. Its a novel finding in Nepal and authors have used adequate advanced techniques for the laboratory experiment. However, only animal and environmental samples are included and human samples are not included in this study. Therefore the role of novel strain of the organism in human infections needs further research work with inclusive of human samples. Language of the manuscript at same places needs revision.

6. PLOS authors have the option to publish the peer review history of their article (what does this mean?). If published, this will include your full peer review and any attached files.

Reviewer #1: No

Reviewer #2: No

---

## [Author Response · Author response to Decision Letter 0]

30 Nov 2022

Reviewer #1: This study reports the novel strains of Campylobacter causing diarrhoeal outbreaks in monkeys in two holy sites of Kathmandu valley.

Major Comment 1 In abstract, the authors have mentioned “Opportunistic diarrheal samples of monkeys were analyzed to ………………” It is better to write diarrhoeal samples only instead of opportunistic samples. It has been replaced with Diarrheal samples.

Major Comment 2 In abstract, your result highlights the detection of Campylobacter in soil and water samples but methodology part is missing. Please mention about collection and processing of soil and water samples We used same protocol to process samples (soil, water), we added the information in line 107.

Major Comment 3 In methods section, line 99, the authors have mentioned water samples were collected form ponds. In contrast to this, in line 102, they have mentioned 5 water samples from river and drinking water sources. Please write exactly which samples you have collected. Also elaborate on the collection and transportation of water samples. We have now clarified the water sample collection process (pond at Swoyambhu /river at Pashupatinath) in line 99 and (pond/river) in line 99.

Major Comment 4 In figure 1, please show the sample collection sites for water and soil samples. This has now been updated.

Minor Comment 5 Please correct the grammatical errors in the text such as

Line 81: Please rearrange the sentence.

 The sentence has been restructured.

Minor Comment 6 Line 100: Please mention the name of the laboratory instead of our lab.

 Our lab has been replaced with the Intrepid Nepal Lab. In line 101. 

Minor Comment 7 Line 111: Please write the symbol µ instead of u

 u has been replaced with symbol µ.

Minor Comment 8 Line 111-113: Please complete the sentence “ The cycling…….4°C.” The sentence has been completed.

---

## [Editor Report · Decision Letter 1]

28 Dec 2022

PONE-D-22-17357R1Novel strains of Campylobacter cause diarrheal outbreak in Rhesus macaques (Macaca mulatta) of Kathmandu ValleyPLOS ONE

Dear Dr. Karmacharya,

Thank you for submitting your manuscript to PLOS ONE. After careful consideration, we feel that it has merit but does not fully meet PLOS ONE’s publication criteria as it currently stands. Therefore, we invite you to submit a revised version of the manuscript that addresses the points raised during the review process.

We look forward to receiving your revised manuscript.

Kind regards,

Dwij Raj Bhatta, PhD

Academic Editor

PLOS ONE

Journal Requirements:

Additional Editor Comments:

1. Please add a paragraph in introduction section /elaborate, by mentioning name of all species of a genus Capylobacter as per official nomenclature &mention determinative morphological biochemical characteristics, host range and medical importance of all species

2. In introduction part: briefly mention types of standard cultural methods and biochemical identification, confirmation of these microaerophilic bacteria with proper reference!

3. Whether the molecular methods applied in the present research is enough to claim novel species? Why cultural methods not applied first to identify isolates and followed by Sequencing of such isolates
---

## [Author Response · Author response to Decision Letter 1]

30 Dec 2022

Reviewers’ comments 

Reviewer #1: This study reports the novel strains of Campylobacter causing diarrhoeal outbreaks in monkeys in two holy sites of Kathmandu valley.

Major Comment 1 In abstract, the authors have mentioned “Opportunistic diarrheal samples of monkeys were analyzed to ………………” It is better to write diarrhoeal samples only instead of opportunistic samples. It has been replaced with Diarrheal samples.

Major Comment 2 In abstract, your result highlights the detection of Campylobacter in soil and water samples but methodology part is missing. Please mention about collection and processing of soil and water samples We used same protocol to process samples (soil, water), we added the information in line 107.

Major Comment 3 In methods section, line 99, the authors have mentioned water samples were collected form ponds. In contrast to this, in line 102, they have mentioned 5 water samples from river and drinking water sources. Please write exactly which samples you have collected. Also elaborate on the collection and transportation of water samples. We have now clarified the water sample collection process (pond at Swoyambhu /river at Pashupatinath) in line 99 and (pond/river) in line 99.

Major Comment 4 In figure 1, please show the sample collection sites for water and soil samples. This has now been updated.

Minor Comment 5 Please correct the grammatical errors in the text such as

Line 81: Please rearrange the sentence.

 The sentence has been restructured.

Minor Comment 6 Line 100: Please mention the name of the laboratory instead of our lab.

 Our lab has been replaced with the Intrepid Nepal Lab. In line 101. 

Minor Comment 7 Line 111: Please write the symbol µ instead of u

 u has been replaced with symbol µ.

Minor Comment 8 Line 111-113: Please complete the sentence “ The cycling…….4°C.” The sentence has been completed.

Reviewer #2: The manuscript is well written. Its a novel finding in Nepal and authors have used adequate advanced techniques for the laboratory experiment. However, only animal and environmental samples are included and human samples are not included in this study. Therefore the role of novel strain of the organism in human infections needs further research work with inclusive of human samples. Language of the manuscript at same places needs revision

Response: The language has been updated at some places in the manuscript.

---

## [Editor Report · Decision Letter 2]

3 Jan 2023

PONE-D-22-17357R2Novel strains of Campylobacter cause diarrheal outbreak in Rhesus macaques (Macaca mulatta) of Kathmandu ValleyPLOS ONE

Dear Dr. Karmacharya,

Thank you for submitting your manuscript to PLOS ONE. After careful consideration, we feel that it has merit but does not fully meet PLOS ONE’s publication criteria as it currently stands. Therefore, we invite you to submit a revised version of the manuscript that addresses the points raised during the review process.

We look forward to receiving your revised manuscript.

Kind regards,

Dwij Raj Bhatta, PhD

Academic Editor

PLOS ONE

Journal Requirements:

Additional Editor Comments (if provided):

Need to address specific comment from academic editor made on their revision 1 manuscript previously sent to authors !! Unfortunately they have submitted manuscript again as revision 2 but it does not satisfy editor's comments and questions previously sent!
---

## [Author Response · Author response to Decision Letter 2]

5 Jan 2023

The comments made by the Editor was addressed in our previous submission by making minor changes to the manuscript and reasons for not addressing his suggestions entirely were explained in the rebuttal letter.

---

## [Editor Report · Decision Letter 3]

9 Jan 2023

Novel strains of Campylobacter cause diarrheal outbreak in Rhesus macaques (Macaca mulatta) of Kathmandu Valley

PONE-D-22-17357R3

Dear Dr. Karmacharya,

We’re pleased to inform you that your manuscript has been judged scientifically suitable for publication and will be formally accepted for publication once it meets all outstanding technical requirements.

Kind regards,

Dwij Raj Bhatta, PhD

Academic Editor

PLOS ONE

Additional Editor Comments (optional):

The reaserch gives important information on campylobacter species that were causative agent of dirrhoea in monkeys in Nepal and confirmed by WGST ! Revision 3 manuscript with track change has been reviewed and rebuttal letter also viewed! ! Revision 3 of the manusctipt be published along with supporting informations and author explanations !
---

## [Editor Report · Acceptance letter]

16 Feb 2023

PONE-D-22-17357R3 

Novel strains of *Campylobacter* cause diarrheal outbreak in Rhesus macaques *(Macaca mulatta)* of Kathmandu Valley 

Dear Dr. Karmacharya:

I'm pleased to inform you that your manuscript has been deemed suitable for publication in PLOS ONE. Congratulations! Your manuscript is now with our production department. 

Kind regards, 

on behalf of

Professor Dwij Raj Bhatta 

Academic Editor

PLOS ONE